# Natrium Diacid Phosphate-Manganese-Lead Vitroceramics Obtained from Spent Electrodes

**DOI:** 10.3390/ma16052018

**Published:** 2023-02-28

**Authors:** Denisa Cuibus, Simona Rada, Sergiu Macavei, Horatiu Vermesan

**Affiliations:** 1Physics and Chemistry Department, Technical University of Cluj-Napoca, 400020 Cluj-Napoca, Romania; 2National Institute for Research and Development of Isotopic and Molecular Technologies, 400293 Cluj-Napoca, Romania

**Keywords:** car battery, recycling, NaH_2_PO_4_-MnO_2_-PbO_2_-Pb vitroceramics, structure, electrode

## Abstract

NaH_2_PO_4_-MnO_2_-PbO_2_-Pb vitroceramics were studied usinginfrared (IR), ultraviolet-visible (UV-Vis) and electron paramagnetic resonance (EPR) spectroscopies to understand the structural modifications as potential candidates for electrode materials. The electrochemical performances of the NaH_2_PO_4_-MnO_2_-PbO_2_-Pb materials were investigated through measurements of cyclic voltammetry. Analysis of the results indicates that doping with a suitable content of MnO_2_ and NaH_2_PO_4_ removes hydrogen evolution reactions and produces a partial desulphatization of the anodic and cathodic plates of the spent lead acid battery.

## 1. Introduction

Lead acid batteriesare the main secondary electrochemical power source and are used in automotive starting power, electric vehicles, aerospace and military equipment, and in energy storage for solar photovoltaics [1,2]. The main characteristics of lead acid batteries are low-energy density, simple control, low costs, and the fact that they are environmentally friendly [3,4].

In Europe, over 99% of lead acid batteries are collected and recycled [3]. Significant amounts of secondary lead can be obtained through lead-acid battery recycling, and pollution is reduced [4]. In spent batteries, the anodic and cathodic electrodes are composed of PbO_2_/PbSO_4_ and Pb/PbSO_4_. For the cathodic electrode, the degradation of active material and positive grid corrosion are the main failure modes of the lead acid battery.

The problem of irreversible sulphatization is the main cause of failure for anodic (negative) electrodes [5]. PbSO_4_ formation on the surface of the electrode increases the internal resistance and the current density, which results in a faster degradation process and the premature failure of the battery. The sulphatization process of the negative electrode surface is the cause of the early failure of the battery and of its limited life. For the reduction of passivation of the electrode surface, it is necessary to reduce the PbSO_4_ size during the discharge process [6,7].

The main technologies used to recyclethe active mass of the lead acid battery are the pyrometallurgical and hydrometallurgical processes, both separately and in combination. Pyrometallurgical technology is mostly used for the recycling of the spent lead acid battery. This process is complex, high in energy consumption, and is carried out at a high temperature—above 1300 °C. Although varied techniques have been used for the recycling of waste lead acid batteries, these need to be further explored because of the risks of environmental pollution and the poor solubility of lead compounds (for the hydrometallurgical process) [8].

It is widely recognized that the addition of various oxides can improve the overall electrical conductivity of the negative plate, produce charge and discharge currents with the anodic plate of the lead-acid battery, limit the growth of lead sulphate crystals, and thus increase the life cycle of the lead [9,10].

The phosphoric acid was used in the lead paste or as an additive in electrolyte solution to reduce the lead corrosion and the self-discharge rate of the lead acid battery [11]. The main problemwith the paste or the electrolyte was that the capacity of the lead acid battery was reduced. Sulphate additives such as Na_2_SO_4_ cannot improve its electrolyte conductivity or reduce the impedance value, but can be helpful in inhibiting the formation of hydrogen and water loss, as well asin improving the cycle life of the battery [12].

The manganese ions in glasses and vitroceramicsare of interest because they have varied valence states, while the phosphate ions have good ionic conductivity [13].

Thisstudy aims to recycle the anodic and cathodic plates from a spent lead acid battery with high wear, and to optimize them through doping withMnO_2_ and NaH_2_PO_4,_using a low-cost method which causes less pollution. The structure of recycled andNaH_2_PO_4_-MnO_2-_doped materials was investigated using X-ray diffraction (XRD), infrared (IR), ultraviolet-visible (UV-Vis) and electron paramagnetic resonance (EPR) spectroscopies. The objective of this study is also to investigate the electrochemical reactions involved in the recycled materials, such as the hydrogen evolution reaction and the passivation process in the sulfuric acid electrolyte. The effect of the NaH_2_PO_4_ content on the recycled material was studied using cyclic voltammetry.Doping of the recycled material with NaH_2_PO_4_ and MnO_2_ was chosen because they can inhibit the reactions of hydrogen evolution.

## 2. Experiment

The anodic plate of the spent car battery served as a Pb source and the cathodic plate as PbO_2_ source. The anodic and cathodic powders MnO_2_ and NaH_2_PO_4,_ which are fine powders in the xNaH_2_PO_4_·5MnO_2_·(95−x)[4PbO_2_·Pb] composition where x = 0, 5, 10, 15, 20, 30, 40, 50, 60 and 70 mol% NaH_2_PO_4,_wereweighed using an analytical balance. The mixture of powders was placed in a ceramic crucible, which was placed in an electric oven set at a temperature of 1050 °C. After 10 min, the melted powderswere quickly put on a steel plate at room temperature.

Crystalline phases were analyzed using an X-ray diffractometer (Rigaku Corporation, Tokyo, Japan) using a radiation with the wavelength λ = 1.54 Å.

The IR spectra of the samples dispersed in KBr were studied with a JASCO FTIR 6200 spectrometer (Jasco Corporation, Tokyo, Japan) which hada resolution of 2 cm^−1^.

The UV-VIS absorption spectra of the powder samples were measured using a Perkin-Elmer Lambda 45 UV-Vis spectrometer (Perkin-Elmer Corporation, Waltham, MA, USA).

EPR measurements of powder samples were recorded using a Bruker ELEXSYS 500 X-band (9.52 GHz) spectrometer (Bruker Corporation, Billerica, MA, USA).

The electrochemical properties were determined through measurements of cyclic voltammetry, using a PGSTAT 302N potentiostat/galvanostat and NOVA 1.11 software, using an electrochemical cell with three electrodes. The prepared samples were used as working electrodes. The platinum was used as a counter electrode and the calomel electrode had the role of a reference electrode. A solution with a concentration of 38% H_2_SO_4_ (similar to that used in a car battery) was used as an electrolyte.

## 3. Results and Discussion

### 3.1. Analysis of X-ray Diffraction

X-ray diffractograms of the prepared system in the xNaH_2_PO_4_·5MnO_2_·(95−x)[4PbO_2_·Pb] composition, where x = 0, 5, 10, 15, 20, 30, 40, 50, 60 and 70 mol% NaH_2_PO_4,_ are shown in Figure 1a,b. The XRD results indicate a vitroceramic structure for all samples. The presence of the PbO·PbSO_4_ crystalline phase with a monoclinic structure and a MnPO_4_ crystalline phase with a hexagonal structure were evidenced in the vitroceramic structure. The amount of the oxo-sulphated crystalline phase decreased for the sample with 70 mol% NaH_2_PO_4_.

In conclusion, we recommend the vitroceramic with 70 mol% NaH_2_PO_4_ as a new electrode for lead acid batteries because the content of the oxo-sulphated crystalline phase is reduced after the recycling of the spent plates.

### 3.2. Infrared Spectra

The FTIR spectra of the vitroceramics with xNaH_2_PO_4_·5MnO_2_·(95−x)[4PbO_2_·Pb] composition where x = 0–70 mol% NaH_2_PO_4_ are shown in Figure 2a,b. The infrared absorption bands were assigned to the stretching modes of the [PbO_3_], [PbO_4_] and [PbO_6_] units. The IR band centered at 460 cm^−1^ causes the Pb-O-Pb and O-Pb-O deforming vibrations of the [PbO_4_] units. The broader IR bands in the range between 650 and 850 cm^−1^ were assigned to the Pb-O stretching vibrations in the [PbO_n_] pyramidal units (with *n* = 3 and 4). The IR band located at 875 cm^−1^ causes the Pb-O stretching modes in [PbO_6_] structural units [14].

Examination of the IR data shows that some structural changes have occurred in the region between 500 and 1200 cm^−1^ by the addition of varied dopant levels.

For vitroceramics with x ≤ 10 mol% NaH_2_PO_4,_ a tendency for the intensity of the IR bands centered at 600 cm^−1^ and those in the range between 1000 to 1200 cm^−1^ to decrease was observed by adding of higher NaH_2_PO_4_ content in the host matrix. The IR bands centered at 600, 1050 and 1150 cm^−1^ correspond to the S-O elongation vibrations in the sulphate units. This evolution indicates that the amount of sulphate phases was decreased through doping with NaH_2_PO_4_ contents.

The intensity of the IR bands located between 530 and 1200 cm^−1^ increasedthrough the addition of dopant contents up to 15 mol% NaH_2_PO_4_, while its intensity decreased for the samples with x ≤ 30 mol% NaH_2_PO_4_. The intensity of the IR band centered at 875 cm^−1^ reached its maximum value for the samples with x = 5, 20, 30 and 40 mol% NaH_2_PO_4_. This evolution indicates that the excess of oxygen atoms changes the lead geometry from [PbO_n_] units with *n* = 3, 4 to octahedral [PbO_6_] units. The IR bands situated in the range between 900 and 1200 cm^−1^ come from phosphate units. The prominent IR band centered at 1100 cm^−1^ corresponds to the elongation vibrations of the S-O bonds in the sulphate units overlapswith the stretching vibrations of the P-O bonds in the metaphosphate units.

For the vitroceramics with x = 30 mol% NaH_2_PO_4,_ a decreasing trend in the intensity of the IR bands between 530 and 1200 cm^−1^ was observed, but its width was increased. Thissuggests that the degree of polymerization of the lead-phosphate matrix was enriched. The intensity of the IR bands decreases gradually to 70 mol% NaH_2_PO_4_.

These compositional evolutions show that the excess oxygen atoms will be accommodated in the host vitroceramic by the formation of [PbO_6_] octahedrons.

The intensity of the IR bands located at about 530, 990, 1100 and 1175 cm^−1^ was assigned the stretching modes of the phosphate units. Monosodium phosphate shows IR bands at 1000 and 1100 cm^−1^ due to the asymmetric stretching vibrations of NaH-PO_4_ bonds in the trigonal pyramidal units. The major changes are due to the increase in intensity of these IR bands.

In conclusion, analysis of the IR data shows important structural changes in the amount of the sulphate and phosphate units by increasing of the NaH_2_PO_4_ content.

### 3.3. Ultraviolet-Visible Spectroscopy

The UV-Vis spectra of the vitroceramic with xNaH_2_PO_4_·5MnO_2_·(95−x)[4PbO_2_·Pb] composition, where x = 0, 5, 10, 15, 20, 30, 40, 50, 60 and 70 mol% NaH_2_PO_4,_ are shown in Figure 3a,b. The UV band centered at 310 nm was correlated with the contributions of Pb^2+^ ions [15]. The intensity of the UV-Vis bands situated at approximately 420 nm and 490 nm is due to the Mn^2+^ and Mn^3+^ ions, respectively [16,17].

The intensity of the UV-Vis band located at approximately 310 nm reaches a maximum value for the sample with x = 15 mol% NaH_2_PO_4_. Its intensity increases through doping with higher NaH_2_PO_4_ contents. This structural evolution shows that the number of free Pb^+2^ ions decreases because the amount of oxo-sulphated crystalline phase decreases through doping. By increasing the NaH_2_PO_4_ to 15 mol%, a gradual conversion of the lead of ionic to covalent positions can be denoted. The intensity of IR bands centered at approximately 470 and 875 cm^−1^ assigned to the [PbO_4_] and [PbO_6_] units increases at higher dopant levels.

Progressive addition of NaH_2_PO_4_ up to 70 mol% indicates a tendency of the absorption edge to be shifted towards higher wavelengths corresponding to the non-bridging oxygen ions (420–480 nm) [18].

UV-Vis data indicate that the number of Pb^+2^ ions decreases through doping, while the number of non-bridging oxygen and manganese ions was enhanced. The non-bridging oxygen has a stronger affinity for the lead atom than for themanganese atoms in accordance with their electronegativity.

### 3.4. Determination of Optical Gap Energy

The (αhν)^1/2^ and (αhν)^2^ as a function of the hν photon energy for the vitroceramic system with xNaH_2_PO_4_·5MnO_2_·(95−x)[4PbO_2_·Pb] composition where x = 0, 5, 10, 15, 20, 30, 40, 50, 60 and 70 mol% NaH_2_PO_4_are shown in Figure 4a,b. The optical gap energy values Egare calculated through extrapolation of the linear parts of these curves. Figure 4c represents the compositional dependence of the gap energy for direct (*n* = 1/2) and indirect (*n* = 2) transitions. The values of the optical gap energy were found to increase from 2.21 to 2.34 eV for indirect transitions and from 2.21 to 2.60 eV for direct transitions. In all cases, the optical gap energy values were less than 3eV, indicating semiconducting behavior for all the prepared samples.

The plots of the gap energy as a function of compositional evolution, x, show an increase in Eg values for samples with x = 10 and 15 mol% and a decrease with the addition of NaH_2_PO_4_ concentrations up to x = 50 mol%. With higher NaH_2_PO_4_ contents, the gap energy increases again. These nonlinear variations in Eg values show structural changes in the host matrix.

The increase in optical gap energy values with the addition of NaH_2_PO_4_ content may be due to the decrease in the concentration of non-bonding oxygen atoms.

### 3.5. Structural Investigation by EPR Spectroscopy

The EPR technique can distinguish between Mn^+2^ and Mn^+4^ ions due to the difference in the number of fine transitions and the value of giromagnetic factor g [15]. Mn^+4^ ions with 3d^3^ electron configuration and S = 3/2 have three fine transitions and a g value of <2. The Mn^+2^ ions in the 3d^5^ electron configuration with S = 5/2 have five fine transitions, where g = 2. The Mn^+3^ ions with S = 2 cannot be detected paramagnetically due to the possibility of the ground state being singlet and broad splitting in a near-zero magnetic field [19].

The EPR spectra recorded at room temperature of samples prepared in the xNaH_2_PO_4_·5MnO_2_·(95−x)[4PbO_2_·Pb] composition where x = 0, 5, 10, 15, 20, 30, 40, 50, 60 and 70 mol% NaH_2_PO_4_ are shown in Figure 5a,b.

The EPR spectrum of host matrix shows a broad resonance signal centered at about 3950 Gauss, which comes from the resonance line located at the gyromagnetic factor g~2 due to Mn^+2^ ions located in octahedral coordination [20] superimposed with the resonance line centered at g~1.9 corresponding to the Mn^+4^ ions. The highly symmetric tetrahedral sites give a signal at g~2 similar to that of octahedral sites of Mn^+2^ ions [21].

Through doping with NaH_2_PO_4_ a new resonance line appears centered at g~4.3 and the signals located at g~1.9 and g~2 indicates a hyperfine structure, except for the sample with x = 5 mol% NaH_2_PO_4_. The resonance line located at g~4.3 corresponds to the strongly distorted octahedral sites of Mn^+2^ ions [22].The intensity of the signal centered at about g~4.3 reaches a maximum value for the sample with x = 5 mol% NaH_2_PO_4,_ after which its intensity decreases drastically for the sample with 20 mol% NaH_2_PO_4_. For higher NaH_2_PO_4_ concentrations (x ≥ 40 mol%), all resonance lines indicate a well-resolved hyperfine structure.

### 3.6. Cyclic Voltammetric Measurements

The cyclic voltammograms obtained for the prepared samples in the xNaH_2_PO_4_·5MnO_2_·(95−x)[4PbO_2_·Pb] composition after scanning the first cycle are shown in Figure 6. For the sample with 70 mol% NaH_2_PO_4,_ varied scan rates, ranging from 1 to 50 mV/s, were used to record the cyclic voltammogram (Figure 7).

In the anodic region of the positive current density, three waves centered at about 0.28, 0.38 and 0.56 V were identified. These waves are attributed to the PbO_2_/Pb^+2^ (0.28 V), P_2_O_6_^−4^/2PO_3_^−2^ (0.38V) and MnO_4_^−^/MnO_2_redox systems (0.6 V).

In the cathode region with negative current density, a reduction wave was evidenced which corresponds to two superimposed peaks located at about −0.126 V and −0.345 V assigned to the Pb^+2^/PbO and PbSO_4_/Pb redox systems. The cyclic voltammograms showed no hydrogen evolution reactions.

The cyclic voltammogram gives well defined oxidation and reduction peaks at a scan rate of 1 mV/s. An increase in the scan rate shows that the anodic peak potential, Epa, centered at around 0.222 V, shifts towards positive values, while the cathodic peak potential, Epc centered at around−0.228 V, shifts to the direction with negative potential value, revealing a kinetic constraint and a quasi-reversible process [23]. With the electrochemical characteristic signatures, including the peak-to-peak separation, ΔEp is 450 mV and Ipa/Ipc ratio is 1.304 at 1 mV/s.

The cyclic voltammograms scanned after three cycles at different scan rates (Figure 8) show that the degree of irreversibility was increasing and the value of the current density was decreasing by the raising of scan rates.

## 4. Conclusions

Samples in the xNaH_2_PO_4_·5MnO_2_·(95−x)[4PbO_2_·Pb] composition where x = 0, 5, 10, 15, 20, 30, 40, 50, 60 and 70 mol% NaH_2_PO_4_, were prepared by the melt quenching method using as raw materials spent electrodes of the lead acid battery, MnO_2_ and NaH_2_PO_4_ powders.

X-ray patterns indicates vitroceramic structures for all samples. The presence of two crystalline phases, namely PbO·PbSO_4_ and MnPO_4_ crystalline phases, were detected in the XRD data.

Analysis of the IR spectra showsa decrease in the sulphate units, and the formation of [PO_4_] units and [PbOn] structural units as a result of adding of higher NaH_2_PO_4_ contents in the host vitroceramic. The intensity of the UV-Vis band located at 310 nm rises because the number of Pb^+2^ ions was decreased through doping.

For the host vitroceramic, the EPR spectrum shows two broad resonance lines centered at about g~2 and 1.9 due to Mn^+2^ and Mn^+4^ ions, respectively. Through doping with NaH_2_PO_4_ concentration, the signals located at g~1.9 and g~2 indicate a hyperfine structure and a new resonance line centered at g~4.3 appeared.

Incorporation of 70 mol% NaH_2_PO_4_ in the host vitroceramic produces partial desulphatization of the recycled plates and electrochemical performances of the electrode material. By increasing the scan rate, the oxidation peak shifts slightly towards the positive potential, while the reduction peak shifts towards the negative potential.

## Figures and Tables

**Figure 1 materials-16-02018-f001:**
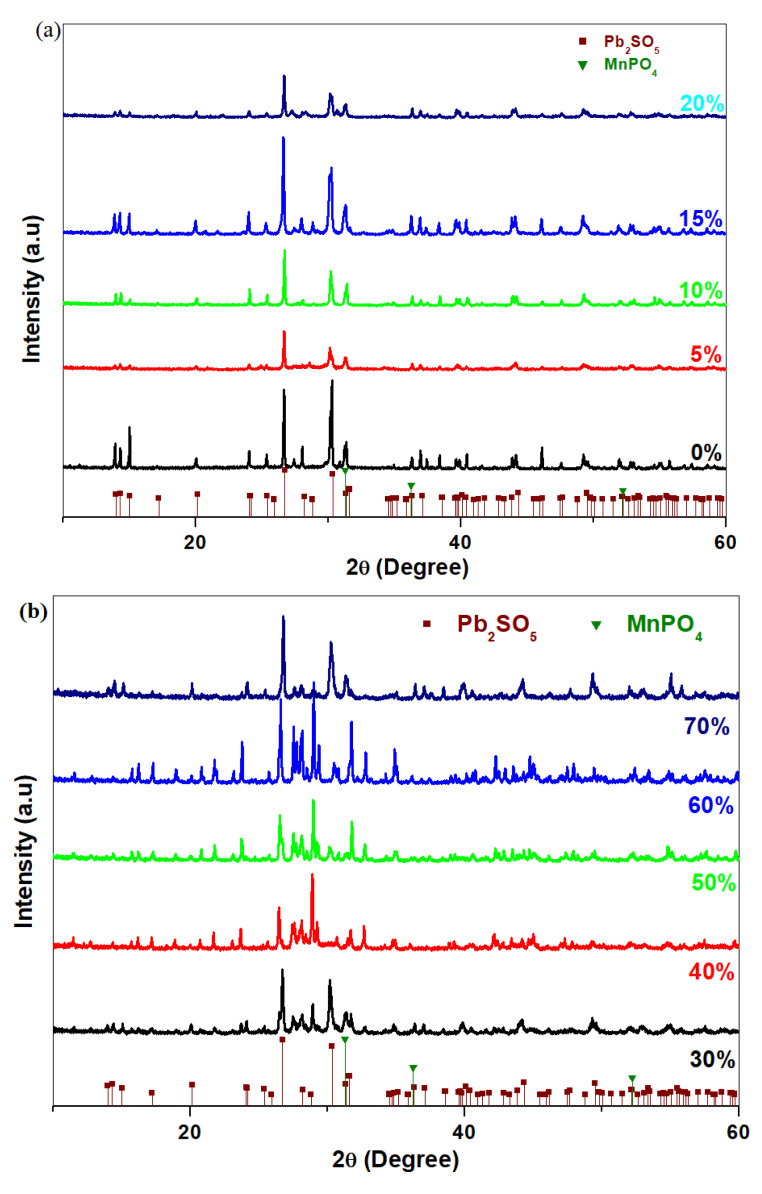
X-ray diffractograms of the vitreous system with the. xNaH_2_PO_4_·5MnO_2_·(95−x)[4PbO_2_·Pb] composition where (**a**) x = 0–20 mol% NaH_2_PO_4_ and (**b**) x = 20–70 mol% NaH_2_PO_4_.

**Figure 2 materials-16-02018-f002:**
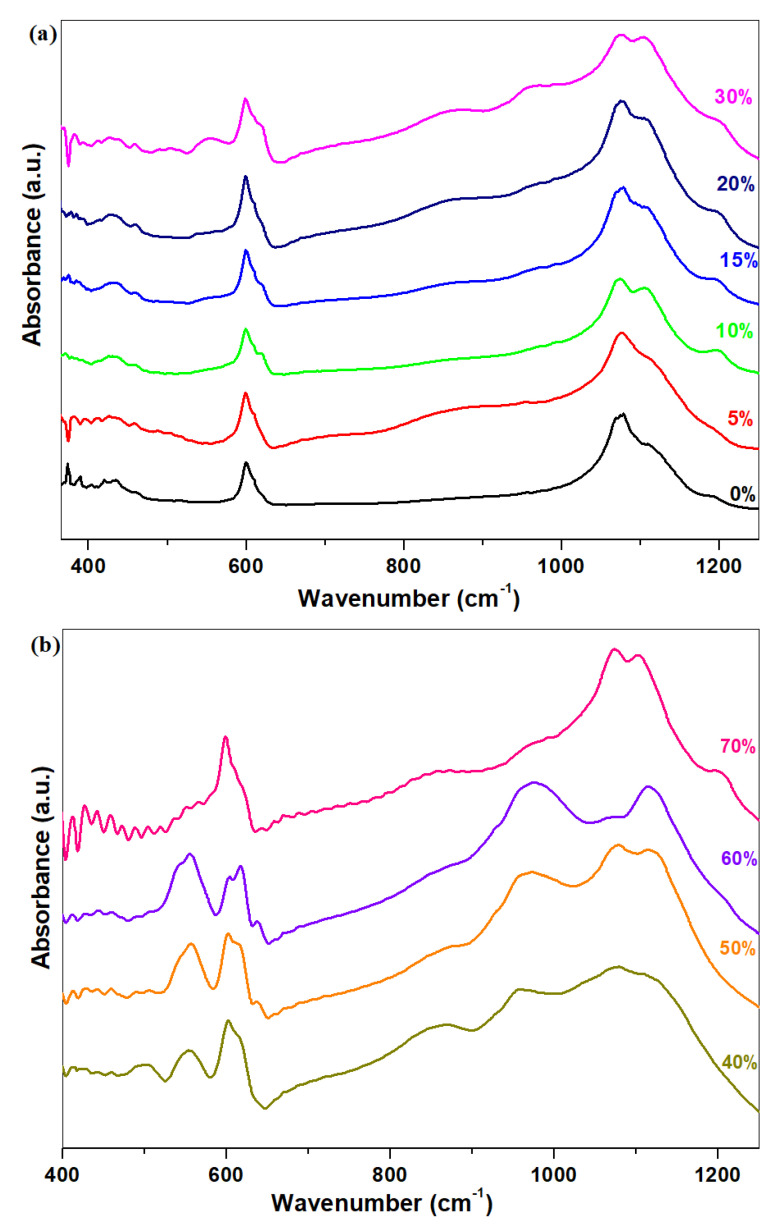
FTIR spectra of the vitreous system with xNaH_2_PO_4_·5MnO_2_·(95−x)[4PbO_2_·Pb] composition where (**a**) x = 0–30 mol% NaH_2_PO_4_, and (**b**) x = 20–70 mol% NaH_2_PO_4_.

**Figure 3 materials-16-02018-f003:**
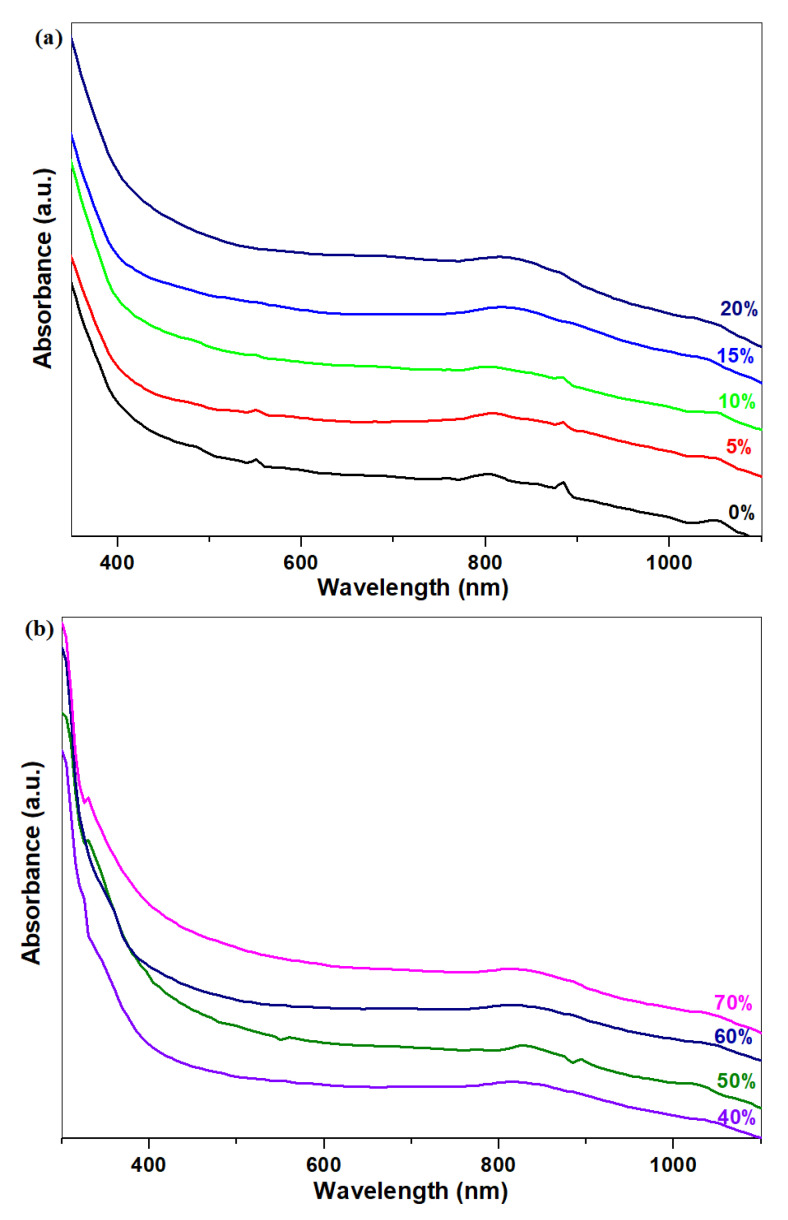
UV-Vis spectra of the vitreous system with the xNaH_2_PO_4_·5MnO_2_·(95−x)[4PbO_2_·Pb] composition where (**a**) x = 0–20 mol% NaH_2_PO_4_ and (**b**) x = 30–70 mol% NaH_2_PO_4_.

**Figure 4 materials-16-02018-f004:**
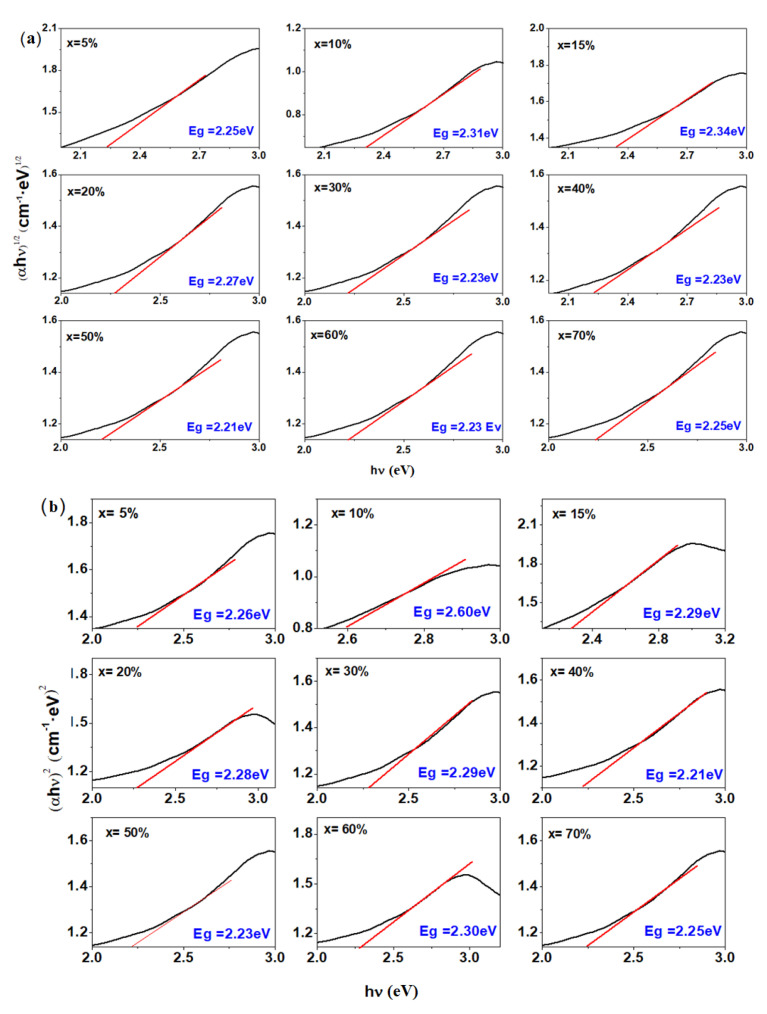
Dependence of (**a**) (αhν)^1/2^ on hν and (**b**) (αhν)^2^ on hν for the vitreous system with. xNaH_2_PO_4_·5MnO_2_·(95−x)[4PbO_2_·Pb] composition where x = 0–70 mol% NaH_2_PO_4_. (**c**) Compositional evolution of the gap energy of direct and indirect transitions.

**Figure 5 materials-16-02018-f005:**
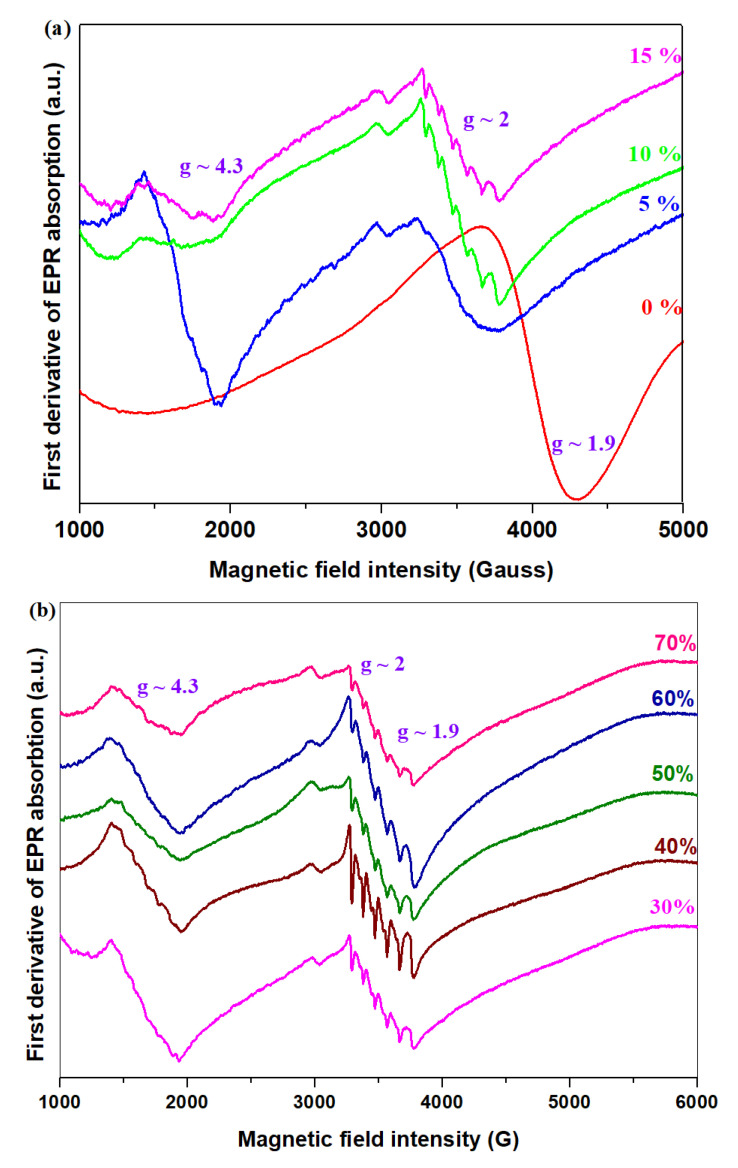
EPR spectra of samples prepared in the xNaH_2_PO_4_·5MnO_2_·(95−x)[4PbO_2_·Pb] composition where (**a**) x = 0–30 mol% NaH_2_PO_4_ and(**b**) x = 40–70 mol% NaH_2_PO_4_.

**Figure 6 materials-16-02018-f006:**
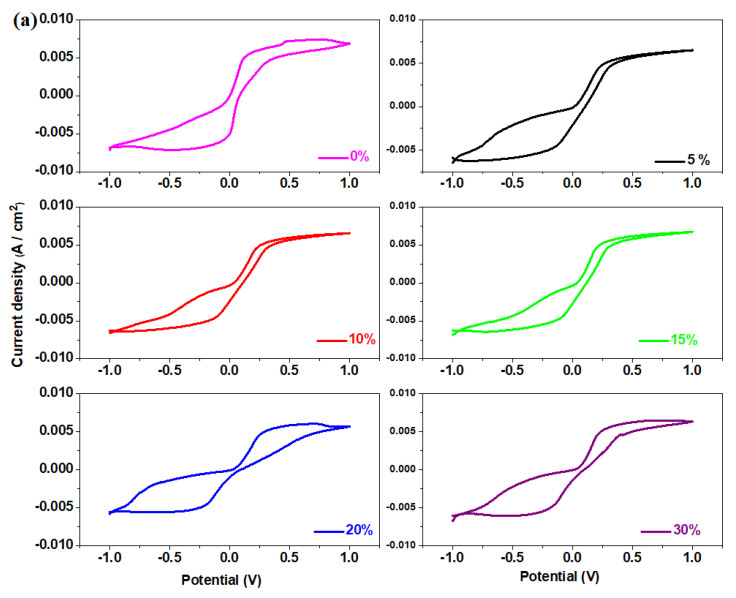
Cyclic voltammogram of the recycled glass system with the xNaH_2_PO_4_·5MnO_2_·(95−x)[4PbO_2_·Pb] composition where (**a**) x = 0–30 mol% NaH_2_PO_4_, (**b**) x = 40–70 mol% NaH_2_PO_4_ and (**c**) x = 0–70 mol% NaH_2_PO_4_.

**Figure 7 materials-16-02018-f007:**
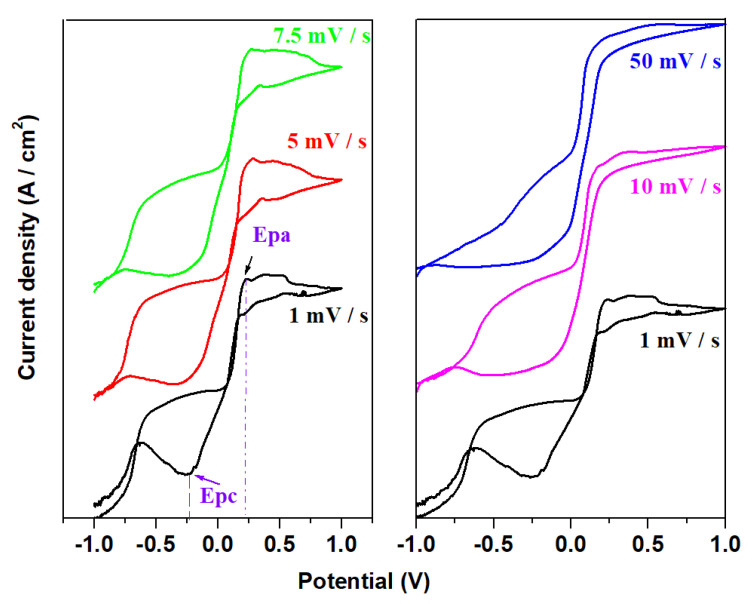
Cyclic voltammogram of the sample with 70 mol% NaH_2_PO_4_ at different scan rates (from 1 mV/s to 50 mV/s).

**Figure 8 materials-16-02018-f008:**
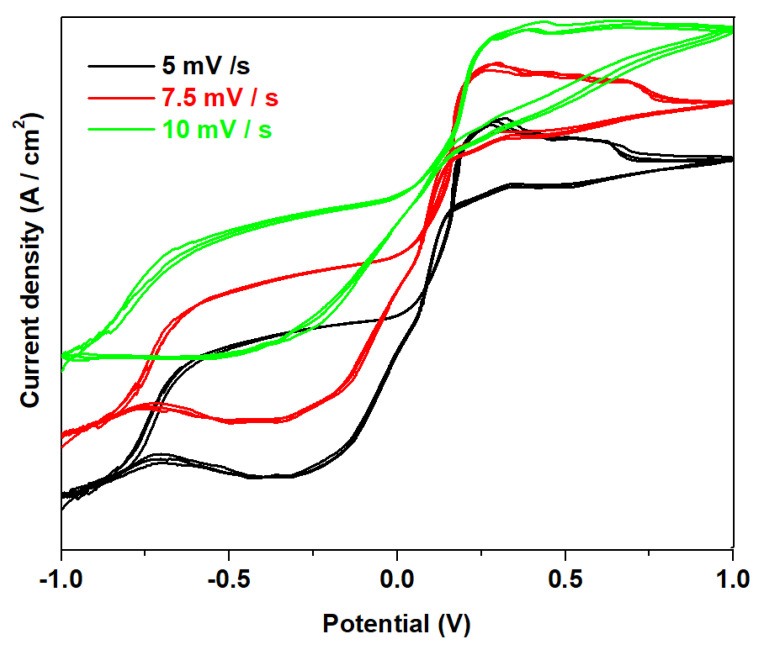
Cyclic voltammograms of the sample with 70 mol% NaH_2_PO_4_scannedafter three cycles at different scan rates (from 1 mV/s to 10 mV/s).

## Data Availability

No new data were created or analyzed in this study. Data sharing is not applicable to this article.

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
