# Peer review of "Natrium Diacid Phosphate-Manganese-Lead Vitroceramics Obtained from Spent Electrodes"

_materials, 2023, doi:10.3390/ma16052018_

Round 1

Reviewer 1 Report

The authors use a combination of spectroscopic techniques such as infrared, UV-vis and EPR to predict potential candidates for electrode materials. The authors recommend the vitroceramic with 70 mol % NaH2PO4 as new candidate for lead acid battery.

I have put my concerns in the attached pdf. To summarize my concerns, the introduction does not provide any background. Please provide a good motivation that includes appropriate and sufficient references etc. Some of the conclusions are not justified. Apart from these issues, the english needs to be modified and needs to be improved.

Author Response

Reviewer 1

The authors use a combination of spectroscopic techniques such as infrared, UV-vis and EPR to predict potential candidates for electrode materials. The authors recommend the vitroceramic with 70 mol % NaH2PO4 as new candidate for lead acid battery.

I have put my concerns in the attached pdf. To summarize my concerns, the introduction does not provide any background. Please provide a good motivation that includes appropriate and sufficient references etc. Some of the conclusions are not justified. Apart from these issues, the english needs to be modified and needs to be improved.

Authors

The Introduction section was improved with suitable references. The motivation of this paper was added in the revised manuscript. The conclusions section was rewritten. The English language was checked.

Reviewer 2 Report

No crystals are labeled in Figure 1(b), please improve it.

Why not combine Figure 2 (a) and (b), as x=20 and 30% are used two times in (a) and (b).

In Figure 4 and Figure 6 all the digits on the coordinate axis are written as “x,x”, please consider to revise them into “x.x”, such as 2.0 other than 2,0.

Author Response

Reviewer 2

  1. No crystals are labeled in Figure 1(b), please improve it.

Authors

  1. In Figure 1b the label of the crystalline phases were evidenced.

Reviewer 2

  1. Why not combine Figure 2 (a) and (b), as x=20 and 30% are used two times in (a) and (b).

Authors

  1. 2. In Figure 2b) the samples with x = 20 and 30 % were deleted.

Reviewer 2

  1. 3. In Figure 4 and Figure 6 all the digits on the coordinate axis are written as “x,x”, please consider to revise them into “x.x”, such as 2.0 other than 2,0.

Authors

  1. The coordinate axes were written as “x.x” in the Figures 4 and 6.

Reviewer 3 Report

In this manuscript, the NaH2PO4-MnO2-PbO2-Pb vitroceramics were prepared as potential candidates as electrode materials. The structure and electrochemical performances of the NaH2PO4-MnO2-PbO2-Pb materials were investigated. Therefore, I suggest that this manuscript should be revised before publication. There are some questions to be solved as shown below:

1) The introduction section should be further perfected including research background and novelty of research content.

2) The functional groups should be labeled on the Figure 2.

3) Why is the optical gap energy of X=30% for indirect transition obviously higher other materials in Figure 4c?

4) The cyclic voltammetry properties of electrode materials with different scan rates should be measured and added.

5) Whether the prepared electrode material can be applied to actual lead acid battery?

Author Response

Reviewer 3

In this manuscript, the NaH2PO4-MnO2-PbO2-Pb vitroceramics were prepared as potential candidates as electrode materials. The structure and electrochemical performances of the NaH2PO4-MnO2-PbO2-Pb materials were investigated. Therefore, I suggest that this manuscript should be revised before publication. There are some questions to be solved as shown below:

1) The introduction section should be further perfected including research background and novelty of research content.

Authors

  • The Introduction section was improved in the revised manuscript.

Reviewer 3

2) The functional groups should be labeled on the Figure 2.

Authors

2) The crystalline phases were labeled on the Figure 2.

Reviewer 3

3) Why is the optical gap energy of X=30% for indirect transition obviously higher other materials in Figure 4c?

Authors

3) The optical gap energy value of the 30 % for indirect transition was modified in the revised manuscript.

Reviewer 3

4) The cyclic voltammetry properties of electrode materials with different scan rates should be measured and added.

Authors

4) The cyclic voltammetry with varied scan rates were added in the revised manuscript (Figures 7 and 8).

Reviewer 3

5) Whether the prepared electrode material can be applied to actual lead acid battery?

Authors

5) The electrochemical cell from experimental procedure simulates the operating conditions of the electrode from the lead acid battery.

Round 2

Reviewer 3 Report

The manuscript has been seriously revised according to the reviewers' suggestions. Therefore, I suggest that the revised manuscript can be accepted for publication.